# Poor compliance with school food environment guidelines in elementary schools in Northwest Mexico: A cross-sectional study

Yazmín Hugues[1], Rolando G. Díaz-Zavala[1], Trinidad Quizán-Plata[1], Camila Corvalán[2], Michelle M. Haby[1] *

1 Division of Biological and Health Sciences, Department of Chemical and Biological Sciences, University of Sonora, Hermosillo, México, 2 Institute of Nutrition and Food Technology (INTA), University of Chile, Santiago, Chile

* haby@unimelb.edu.au

**Data Availability Statement:** All relevant data are within the paper and its Supporting Information files.

**Funding:** The Division of Biological and Health Sciences of the University of Sonora funded the

## Abstract

### Background

In Mexico, 35.5% of school-age children were overweight or obese in 2018. The school food environment is important because children spend a significant part of their time at school and consume one-third to one-half of their daily meals there. In 2014, a Federal Government guideline for the sale and distribution of food and beverages in Mexican schools was published (the AGREEMENT) but the extent of its implementation is not known.

### Methods

Descriptive cross-sectional study in a representative, random sample of elementary schools, using the tools of the INFORMAS network. Data collection included: a) an interview with a school authority; b) a checklist of items available in the school canteen; c) a checklist of the school breakfast menu; and d) an evaluation of the physical environment. The main indicators were: percentage of implementation (self-report) of the AGREEMENT and percentage of compliance (researcher verified) with the AGREEMENT (based on tools b and c).

### Results

119 schools participated (response rate 87.5%), with 15.1% (95%CI 9.2–22.8) of the schools reporting having fully implemented the AGREEMENT. However, only 1% (95%CI 0–5.3) of the school canteens and 71.4% (95%CI 57.8–82.7) of the school breakfast menus fully complied with the AGREEMENT. A variety of sugar-sweetened beverages and energy-dense, nutrient poor products were found in the school canteens. Further, only 43.7% of the water fountains in schools were functional and 23.4% were clean. In only 24.4% of schools had the school authorities received formal training related to the AGREEMENT and in 28.6% of schools had the parents received information about the AGREEMENT.

printing of the instruments for data collection as well as fuel costs. YHA received a Masters degree scholarship from CONACYT (National Council of Science and Technology). The funders had no role in study design, data collection and analysis, decision to publish, or preparation of the manuscript.

**Competing interests:** The authors have declared that no competing interests exist.

**Abbreviations:** EDNP, energy-dense, nutrient-poor; SSBs, sugar-sweetened beverages.

## Conclusion

The AGREEMENT has been poorly implemented in elementary schools in Mexico. Actions are needed to encourage and support its full implementation to improve the food environment in Mexican schools.

## Background

Overweight and obesity are a public health problem in Mexico and affect all groups of the population, including children [1]. According to the latest publication from the National Health and Nutrition Survey (ENSANUT 2018), 35.5% of Mexican school age children were overweight or obese in 2018 [2]. The state of Sonora is not an exception, with a prevalence of overweight and obesity of 36.9% in school age children in 2012 [3], which was above the national average.

Increased consumption of sugar-sweetened beverages (SSBs) among children is associated with higher caloric intake [4] and there is increasing and stronger evidence that consumption of SSBs is a risk factor for obesity and other health complications [5, 6]. There is also evidence that children's energy excess comes predominantly from processed foods with high levels of cholesterol, saturated fats, sugar, and sodium [7], denominated energy-dense, nutrient-poor (EDNP) foods. The intake of SSBs and EDNP foods is high in Mexico [1].

For children, the school food environment is particularly important because they spend an important part of their time in it [8, 9] and consume one-third to one-half of their daily meals at school, making this a crucial setting for interventions to promote a healthier diet [10, 11]. Policies designed to influence the school food environment have been successful in changing dietary behaviors [10] and are key to improving it [12]. In recognition of this, in 2010 the Mexican government established the general guidelines for dispensing and distribution of foods and beverages at school food establishments in Mexican schools, with the objective of stopping the epidemic of overweight and obesity [7]. However, an evaluation of the implementation of the guidelines in 2012 showed a lack of improvement of the nutritional content of foods and beverages in schools [7, 9].

In 2014 an updated version of the national guideline–AGREEMENT that establishes the general guidelines for the sale and distribution of prepared and processed foods and beverages in the schools of the National Education System–was implemented [13]. However, the extent of its implementation is not known. To date, only a small but in-depth study by the consumer organization El Poder del Consumidor has been conducted, which suggested that there were problems with the AGREEMENT's implementation [14].

In recent years, Mexico has implemented a number of policies to prevent obesity, including taxes on processed foods and SSBs [15, 16], and in October 2020 implemented a new front-of-package labelling system that includes warning signs [17]. Knowing the extent of implementation of the "AGREEMENT" is important for guiding efforts targeted to improvement of the school food environment.

The aim of the current research was to describe the implementation of, and compliance with, the 2014 version of the AGREEMENT in a representative sample of elementary schools in Hermosillo, Sonora, Mexico. Hermosillo is the capital of the northwestern Mexican state of Sonora and had a population of 852,000 in 2019. Together, with the central region of Mexico, school age children in the northern region tended to have a greater consumption of snacks, sweets and desserts than other regions of Mexico in 2016, though no difference in consumption of SSBs, which is high in all regions (79–84%) [1].

## The 2014 AGREEMENT

This AGREEMENT is the main guideline for the regulation of food and beverages available in Mexican schools [13]. The AGREEMENT was published in the Official Gazette of the Federation (Diario Oficial de la Federación) as a guideline and is applicable to both public and private schools in Mexico. It gives responsibility to both the education and health authorities to disseminate, implement and monitor compliance with the guidelines. It also allows for administrative sanctions and fines for the educational authorities under the General Education Law in the case of infractions. However, to date there have been no known reports of any type of penalty for non-compliance or non-implementation of the AGREEMENT in Mexican schools. Further, while the 2010 version of the guidelines specifically states that the intention was for them to be a legal instrument of compulsory observance in schools throughout the country, there is no such statement in the 2014 version of the guidelines (the AGREEMENT) [13, 18].

The AGREEMENT specifies the nutritional characteristics for foods and beverages dispensed at any school food establishment [13]–and includes both food group-based and nutrient-based guidelines. In addition, it establishes that there must be policies that prohibit the sale of food and beverages that do not meet the nutritional criteria in the AGREEMENT. The AGREEMENT applies to all school levels (kindergarten, elementary, secondary, high school and college/university). Among the most important criteria, the AGREEMENT requires that elementary school food establishments:

- Only offer natural foods from Monday to Thursday

- Do not sell sodas any day of the week

- Junk food (EDNP foods) and SSBs that meet certain criteria can only be sold on Fridays

  Some of the other criteria included in the AGREEMENT are:

- The school must have a Committee for School Food Consumption Establishments

- The Committee for School Food Consumption Establishments must convene parents to participate in actions related to the sale and distribution of food and beverages in the school

- Education authorities should coordinate with the government to regulate mobile food vendors outside of schools

- Food and/or beverage providers must comply with the criteria of the AGREEMENT

- The education authorities should disseminate the content of the AGREEMENT

- The education authorities guarantee that schools have drinkable water and infrastructure for proper food hygiene

- The education and health authorities should provide information and training to the Committee of School Food Consumption Establishments, food providers and parents

## INFORMAS

The study was conducted as part of the International Network for Food and Obesity/Noncommunicable Diseases Research, Monitoring and Action Support (INFORMAS) framework. INFORMAS was founded in 2013 to monitor and benchmark the healthiness of national food environments and policies in a standardized way [19]. The framework consists of nine modules covering different aspects of food environments and related policies. This study falls

under the food provision module [20] that seeks to answer the question: What is the nutritional quality of foods and nonalcoholic beverages provided in different settings (e.g. schools, hospitals, workplaces)? It involves, as a minimum, the measurement of two main indicators: 1) % of schools or other publicly funded institutions that implemented the policy or programme; and 2) % of schools or other publicly funded institutions complying with the policy or programme [20].

## Methods

This was a descriptive, cross-sectional study, representative of all elementary schools in Hermosillo, Sonora, Mexico. The school food environment was examined in terms of foods provided and sold with reference to the requirements of the AGREEMENT and the framework and indicators of INFORMAS [20].

The protocol for the study was approved by the Bioethics Committee of the Department of Medicine and Health Sciences, University of Sonora. Permission to contact schools was obtained from the Ministry of Education, who also sent a letter to each of the schools to inform them about the project and to introduce the researchers so that they would be allowed entry to the school. Additionally, at the beginning of each visit a paper copy of the informed consent form was given to the school authority (usually the school principal). They were given time to read it and to ask any questions about the implications of their school's participation in the study. The informed consent form had to be signed by the school authority (written consent), the responsible researcher for the project and a witness before including the school in the study.

### Data collection

Data collection included collection of general data about the school; an interview with a school authority; analysis of foods sold in the school canteen; analysis of the school breakfast menus; and an evaluation of the physical environment. The instruments used (S1 File) were developed specifically for this study because the INFORMAS methodology does not include specific instruments given that these will need to be adapted to the specific country context and food policies. English versions of the instruments are provided in S2 File.

**Interview with a school authority.** A school authority from each school was interviewed, the interview was conducted using a semi-structured questionnaire (S1 File, Part 2). The person interviewed could be the school principal or some other school authority with sufficient knowledge of the school's policies (as determined by each school). The instrument was adapted from a tool developed by Erica D'Souza and colleagues as part of the New Zealand project of INFORMAS [21, 22] and is based on the wording and intent of the AGREEMENT. This instrument sought to obtain information about the level of implementation of the AGREEMENT and to identify barriers and facilitators to its implementation. It included both closed and open questions. Two data collectors were involved. While one asked the questions, the other took notes so as not to lose the details of the conversation. The interview took approximately 20–40 minutes to complete.

**School canteens.** The instrument developed for the collection of information in school canteens included a list of items that are regularly available in school canteens, including items specific to Sonora [23] (S1 File, Part 3). The food and beverage items included were classified into three different categories based on their nutritional status and resulting level of restriction under the AGREEMENT (see Table 1). This approach to school canteen menu assessment has been used by other researchers evaluating the implementation of school food policies [24, 25]. Completion of the checklist was based on observation of items on display and by asking the

**Table 1. Food classification for school canteens, using the traffic light system, according to the 2014 AGREEMENT.**

| Category | Description | Examples |
|---|---|---|
| **RED** | Foods or beverages **restricted** for sale according to the AGREEMENT. These canteen items are very low in nutritional value and are high in saturated fat and/or added sugar and/or added salt. Some of these foods can be sold on Fridays only if they meet specific nutritional criteria of the AGREEMENT. | Cookies, pastries, candies, desserts, salty snacks (potato chips and other processed foods), seeds and nuts with added salt, high in salt/fat snack cheeses, sodas or other SSBs, sandwiches with no vegetables included on white bread, quesadilla (tortilla with melted cheese) made with flour tortillas and instant soup. |
| **AMBER** | Foods or beverages that should be **selected carefully**. These items are moderate in added fat and/or sugar and contribute to excessive calorie intake and, according to the AGREEMENT, this food could be sold occasionally i.e. no more than 2 times per week. | 100% natural fruit/vegetable juices with no added sugar, milk with non-caloric sweeteners and soy beverages with non-caloric sweeteners, horchata, jamaica and other "aguas frescas" without added sugar. |
| **GREEN** | Foods or beverages **recommended** by the AGREEMENT. These items are high in nutrients and fiber and are low in saturated fat and/or added sugar and/or salt. These foods can be sold any day of the week. | Fresh fruits and vegetables, milk without added sugar, whole grain cereals, seeds and nuts without added salt, low-fat cheeses, simple water, sandwiches with vegetables on wholegrain bread, quesadilla with corn tortilla and homemade soup with vegetables. |

SSBs–sugar-sweetened beverages.

vendors about foods that they usually sold but were not in the school canteen at the time of the visit. To ensure complete data, this checklist was completed just prior to the first break (recreation) period in all schools.

**School breakfasts.** The instrument was similar to the school canteens instrument and used the same color classification for foods. Information was collected from an official copy of the menu in schools that were participating in the government school breakfast program or other private or social breakfast programs. The data collectors also spoke with the person responsible for the program (including the cook, if possible) to obtain extra data regarding food preparation, i.e. changes of ingredients, addition of sugar, etc. The data collectors marked all the items that were included in the menu or were used for the food preparation (S1 File, Part 4).

**Evaluation of the physical environment.** This instrument was used to collect information about the availability of drinking water, the number and status of water fountains, and the presence of promotional material outside of the classrooms in the school grounds as well as the number of mobile food vendors that were observed outside of the school (where possible) (S1 File, Part 5).

**Procedure.** The school visits were conducted in pairs by a team of four Nutritionists (YHA and nutrition interns as part of their social service). Prior to data collection, YHA trained all three students, considering the following main aspects: 1) general understanding of INFORMAS methodology and its food provision module, 2) review of previous studies of the food environment in Mexican schools and, 3) review and practice in the use of the instruments for the study (S1 File).

Data was collected between November 2018 and April 2019 in both morning and afternoon shifts. In Mexico, public schools generally have two shifts to allow each school to accommodate double the number of students. To harmonize the data collection, school visits in the first week were made by all four data collectors together. From the second week onwards, visits

were conducted in pairs but YHA participated in the visits of both shifts as a supervisor (for approximately one month) until the data collectors were sufficiently prepared to work without supervision. The data collectors used a checklist that included the step by step instructions for the visit to the school (S1 File, Part 7). The checklist had to be fully completed at the end of the visit to guarantee that the visit was complete.

The data collection was limited to Monday to Thursday as these are the days when all the restrictions apply under the AGREEMENT. The restrictions are more liberal on Fridays and full implementation of the AGREEMENT would be much more complex to evaluate on this day.

Prior to data collection the procedure and instruments were tested in a small group of schools (n = 5) to assess the need for adjustments to the tools. However, no adjustments were needed so the data collection for the definitive study was continued and data from these 5 schools were included in the sample.

In those schools that did not accept to participate, the data collectors sought answers to three of the principal questions from the interview instrument to allow a comparison of responders and non-responders to check for possible selection bias. This short questionnaire was called a "Non-participation survey" (S1 File, Part 6).

## Indicators

The main indicators for this study were based on the INFORMAS framework and include [20]:

Percentage (%) of schools that implemented the AGREEMENT.

Percentage (%) of schools complying with the AGREEMENT.

These indicators are not defined in the INFORMAS framework but the general interpretation of implemented is that a regulation or policy is in place; while compliance implies an assessment of, or evidence that, the regulation or policy that is in place is being followed. Therefore, for this study the concept "implementation" was defined as the practices related to putting the AGREEMENT into practice–as reported by the school authority in the interview [20, 26]. The concept "compliance" was defined as the level to which the sale and distribution of foods within the school matched the specifications in the AGREEMENT [20]. It was evaluated based on observation by the researchers, categorization and overall assessment of the percent of products in each category in the school canteens and school breakfast menus at the time of the visit.

The AGREEMENT was classified as "fully implemented" when the interviewed school authority reported that the implementation of the AGREEMENT in the school was complete or almost complete; as "partially implemented" when the interviewed school authority reported the implementation of the AGREEMENT in the school had already begun but they were still working to complete it; or as "not implemented".

The schools were categorized as showing "full compliance" when 100% (or close to 100%) of the school canteen items or menu items were from the 'green' or 'amber' classification; and as "partial compliance" when at least 50% of the school canteen items or the menu items were from the 'green' or 'amber' classification. This categorization has been used by others to assess compliance with a healthy school canteen policy [26] and the assessment method has been reported to have good validity [27].

## Sample size calculation and selection of schools

A random sample of elementary schools of Hermosillo was selected using a list of random numbers generated in Excel. The list of schools was provided by the Ministry of Education and

includes all public and private schools. For the sample size calculation the formula corresponding to descriptive studies with finite populations was used [28]. A confidence level of 95% (Z = 1.96), a 25% expected prevalence (p = 0.25, q = 0.75) and an accuracy of 5% were used, giving a sample size of 150 schools. However, a lower than 25% prevalence for the key indicators (as found in this study) would reduce the sample size needed or, for the same sample size, result in a narrower confidence interval.

### Data management and analysis

The collected data were entered into a Microsoft Office Excel 2013 spreadsheet. All data were double-checked at both collection and data entry stages. The data analysis was conducted using the software StataSE® v14 for Windows. The 95% confidence intervals for prevalences were calculated using the exact method because one main indicator had a prevalence of less than 5%, which precludes the use of other readily available approximations [29]. Proportions were compared using Fisher's exact test. Missing data are reported and removed from the denominator.

The answers to the open questions from the interview were also transcribed into an excel sheet, preserving the original phrases of the school authorities. Question 18 was an open question that asked: "What do you think can be done to address the barriers to implementation? At the end of the questionnaire there was also a comments section where the data collectors noted opinions and extra comments offered by the person interviewed. This information was classified into themes and subthemes, taking into account the recommendations of Krefting for qualitative data analysis [30]. The number of mentions of each theme was counted and themes were prioritized depending on their frequency to reflect the school authorities perspective about barriers to implementation of the AGREEMENT [31].

## Results

The first 142 schools randomly selected were approached to participate in the study. Six of these schools were found to no longer be in operation so were removed from the sample. Data was collected from 119 schools, giving a response rate of 87.5% (119/136). Of the 119 participating schools, 74.8% were public schools and 30.3% operated in the afternoon shift. The response rate in private schools (66.7%) was considerably lower than that in public schools (97.8%). The median number of students per school was 237 (interquartile range 140–347, minimum 8 and maximum 756).

In 89.9% of schools, the school principal participated in the interview, while the remaining interviewees were teachers (4.2%) or other members of the administrative staff (5.9%). Visits were made to 104 school canteens in 103 schools. One school had two canteens but offered the same products in both, so the data was analyzed together as one canteen, giving 103 in total. 16 schools did not have a canteen or any other formal/informal food establishment inside the premises. Breakfast programs were operating in 58 schools, of which 56 provided a copy of the menu and were included in the data analysis.

Of the 17 schools that did not accept to participate in the study, 13 completed the non-participation survey (S1 File, Part 6). When the results of the 13 non-participant schools were compared to the 119 schools that participated in the study it was difficult to assess the possibility of selection bias in our results. While the participating and non-participating schools were not significantly different in relation to reporting having healthy food policies (47.9% and 53.8%, respectively, P = 0.06), the participating schools were more likely to report not having enough water fountains (30.3% vs 0%, P = 0.01) and more likely to report having received information about the AGREEMENT (68.9% vs 30.8%, P = 0.009). These differences could be partly explained by the fact that 92.3% of the non-participant schools were private schools.

## Implementation and compliance with the AGREEMENT

Despite the fact that the first version of the AGREEMENT was published in 2010, the interview with the school authorities showed that only 15.1% (95%CI 9.2–22.8) of schools reported having fully implemented the AGREEMENT and 55.5% (95%CI 46.1–64.6) had partially implemented it (Fig 1). The above is consistent with the fact that only 24.4% of the school authorities had received formal training related to the AGREEMENT, and only 1.7% of them were able to show an official copy of the document. Full implementation of the AGREEMENT was higher in private schools than in public schools (33.3 vs 9.0%, respectively, p = 0.009, Fisher's exact test). While the AGREEMENT requires that schools have a Committee of School Food Consumption (made up of parents and guardians) to regulate the sale and distribution of food and beverages [13], only 36.1% of the schools reported having a committee. While 65% of schools had received verification visits from the Ministries of Health, Education or both to check the compliance of their school canteen with the AGREEMENT, the proportion of public schools that reported having had a verification visit was significantly higher than in private schools (73.0 vs 39.9%, p = 0.003, Fisher's exact test). More results of the interview can be found in Table 2.

Regarding compliance with the AGREEMENT, only 1% (95%CI 0–5.3) of the school canteens fully complied with it and 3.9% (95%CI 1.1–9.6) partially complied with it (Fig 1). A variety of foods and beverages was found in school canteens (S1 Table). The most frequent items available were: cookies, cakes, candies and sweets (found in 98% of the school canteens); bottled water (92.2%); tacos and "burritos" using a wheat flour tortilla (91.3%); processed juices and nectars (91.3%); fresh fruit and vegetables (87.4%); and dried legumes with added salt (87.4%). At least one type of SSB (e.g. iced tea, sodas, milk with added sugar, nectars and juices) was observed in 100% of the school canteens.

Compliance with the AGREEMENT for the school breakfast menus was much higher, with 71.4% (95%CI 57.8–82.7) of the menus fully complying and 19.6% (95%CI 10.2–32.4) partially compliant (Fig 1). Of the 56 schools assessed, 82.1% were part of the School Breakfast Program (operated by the National System for the Integral Development of the Family–DIF). The 56 menus were identified as "cold breakfasts" (that consist in whole cereal cookies, a carton of whole milk and occasionally dried fruit), "hot breakfasts" (that includes prepared hot meals such as scrambled eggs, sandwiches or pasta), or "mixed breakfasts" (a mix of both hot and cold). Given that the cold breakfasts represent the greatest proportion of the menus analyzed (53.6%), it was expected that their components would be some of the most popular items:

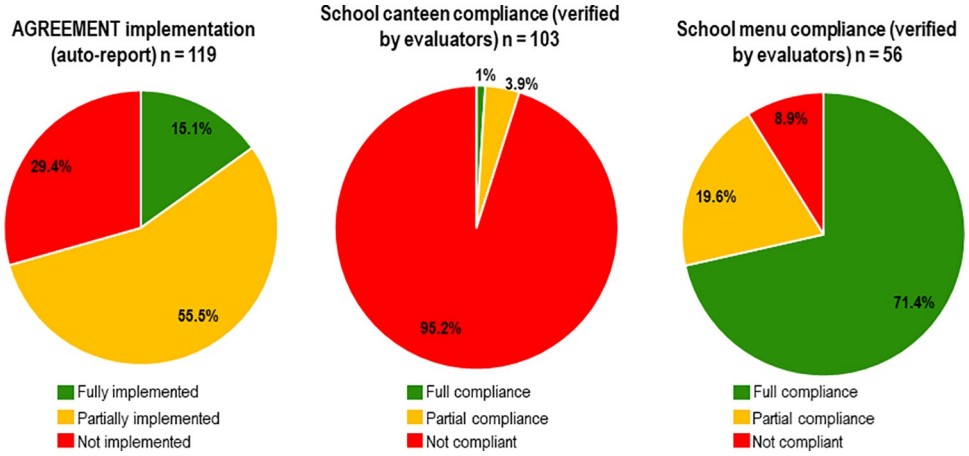

**Fig 1. Level of implementation and compliance with the AGREEMENT in elementary schools of Hermosillo.**

**Table 2. Results of the interview with the school authorities (n = 119).**

| Indicator | Prevalence % | (95% CI) |
|---|---|---|
| Schools where the school authorities had heard about the AGREEMENT before the interview (Q21) | 73.1 | (64.2–80.8) |
| Schools where the school authorities had received information about the AGREEMENT (Q14) | 68.9 | (59.8–77.1) |
| Schools where principals had access to a copy (printed or digital) of the AGREEMENT (Q22) | 11.8 | (6.6–19.0) |
| Schools where it was verified that principals had a copy (printed or digital) of the AGREEMENT | 1.7 | (0.2–5.9) |
| Schools where the teachers had access to a copy (printed or digital) of the AGREEMENT (Q24) | 16 | (10.0–23.8) |
| Schools where the school authorities had received formal training related to the AGREEMENT (Q15): | 24.4 | (16.8–33.2) |
| From SEC | 22.6 | (15.3–31.3) |
| From their boss or superior | 1.7 | (0.2–6.1) |
| Schools that had a committee that regulates the sale and distribution of food and beverages (Q27) | 36.1 | (15.5–31.3) |
| Schools where the parents received information related to the AGREEMENT (Q26) | 28.6 | (20.7–37.6) |
| Schools where there are promotional materials related to the AGREEMENT (e.g. good eating habits, fruit consumption, etc.) (Q28) | 20.2 | (13.7–29.2) |
| Schools that have policies related to junk food or sugar sweetened beverages (Q10) | 47.9 | (38.7–57.2) |
| Schools that allow junk food or sugar sweetened beverages at festivals or fetes | 31.1 | (22.9–40.2) |
| Schools that allow junk food or sugar sweetened beverages at birthday parties | 19.3 | (12.7–27.6) |
| Schools that allow junk food or sugar sweetened beverages on festive or special days | 39.5 | (30.7–48.9) |
| Schools that allow junk food or sugar sweetened beverages every day | 0.8 | (0.0–04.6) |
| Schools that have received verification visits to the school canteen from an external authority (Q13): | 65 | (55.6–73.5) |
| From SS | 29.1 | (21.0–38.2) |
| From SEC | 46.2 | (36.9–55.6) |
| From both SS and SEC | 12.8 | (7.4–20.3) |
| Schools that have raised funds by selling food and/or beverages (Q5): | 74.8 | (66.0–82.3) |
| Chocolate bars | 10.9 | (5.9–18.0) |
| Cakes, pies or cookies | 33.6 | (25.2–42.8) |
| Potato or corn chips | 36.1 | (27.5–45.4) |
| Prepared food (i.e. Mexican food*) | 68.1 | (58.9–76.3) |
| Schools that receive a percentage of the income from the sale of foods and beverages (Q12) | 97.2 | (92.0–99.4) |
| Schools where there was at least one place (inside or outside the school) where children regularly buy food or beverages (Q4): | 89.1 | (82.0–94.1) |
| The school dining room | 21.8 | (14.8–30.4) |
| The school canteen | 85.7 | (78.1–91.5) |
| A vending machine | 0.8 | (0.0–4.6) |
| From mobile food vendors | 60.5 | (51.1–69.3) |
| From teachers or administrative staff | 0.8 | (0.0–4.6) |
| School authorities that reported that the school has some kind of control of mobile food vendors outside of the school (Q6) | 46.2 | (37.0–55.6) |
| School authorities that consider that the school does not have enough water fountains (Q9), because: | 30.3 | (22.2–39.3) |
| The students have to buy bottled water | 16.8 | (10.6–24.8) |
| The water fountains are not within reach of the students | 2.5 | (0.5–7.2) |

(*Continued*)

**Table 2.** (Continued)

| Indicator | Prevalence % | (95% CI) |
|---|---|---|
| The water fountains are dirty or do not function very well | 16 | (10.0–23.8) |

CI–confidence interval; Q–question number; SS—Ministry of Health; SEC—Ministry of Public Education.

*Mexican food: Among the staples of traditional Mexican food are beans, chili and corn. Fried and stewed food predominates in many of its dishes.

whole grain cereals (found in the 89.3% of the menus), milk without added sugar (75%), nuts and seeds (67.9%), dried fruit (66.1%), fresh fruits (50%) and vegetables (46.4%). Full details of the foods included in the menus can be found in S2 Table.

## Barriers and facilitators for the AGREEMENT

In relation to groups or individuals that were barriers to the implementation of the AGREE-MENT in schools, the school authorities considered that parents and students are the principal barriers, followed by food vendors (Fig 2). Parents, students and vendors were more likely to be reported as barriers for public schools than for private schools. Of the school authorities that selected parents and/or students as a barrier, the most common reason was the "lack of knowledge of the AGREEMENT" (70.5%, 95%CI 59.8–79.7), followed by "lack of interest" (69.3%, 95%CI 58.6–78.7). The perception of the school authorities interviewed regarding barriers to implementation of the AGREEMENT and the most frequent possible solutions that they mentioned are shown in Table 3.

In relation to facilitators, teachers and school authorities were reported as the main facilitators of the AGREEMENT, followed very closely by external authorities (such as the municipal government, education and health authorities) (Fig 2). Contrary to that expected, both parents and students were also identified as facilitators to the implementation of the AGREEMENT. Although the classification of facilitators and barriers was determined by two different questions (with the same answer options), many school authorities verbally commented that they also selected parents as facilitators because, with the appropriate orientation and support,

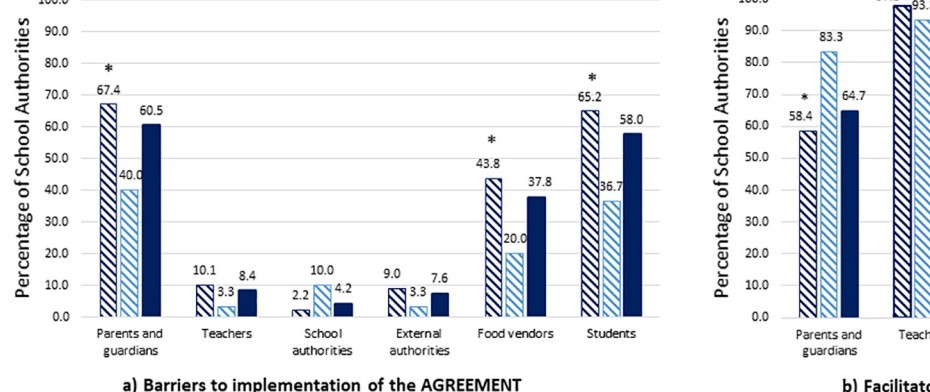
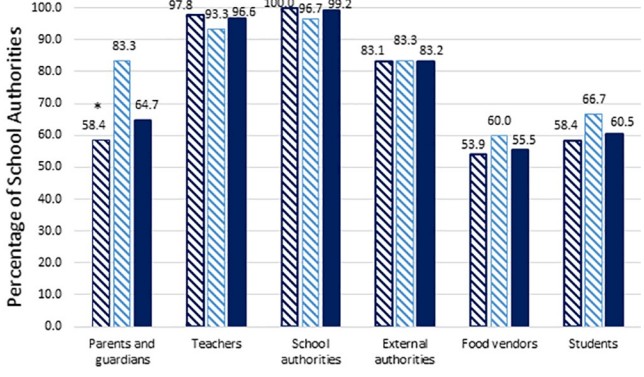

* P<0.05 for difference between public and private schools, Fisher's exact test.

**Fig 2. Groups or individuals that are a) barriers to or b) facilitators for implementation of the AGREEMENT–reported by the school authorities (n = 119).**

**Table 3. Perception of the barriers and potential solutions mentioned by the school authorities during the interview (n = 119).**

| Theme | No. of mentions | Quotes about possible solutions |
|---|---|---|
| Lack of training for parents and/or teachers and/or students | 44 | *"Conduct projects, talks and workshops for children and parents"* |
| | | *"Disseminate the AGREEMENT to parents"* |
| | | *"Educate parents, but it is very difficult"* |
| | | *"Educate/train the schools and parents and give information to the people in general, giving support for access to a healthier diet"* |
| | | *"To conduct information sessions for parents, include talks about the AGREEMENT as part of the school rules"* |
| Lack of awareness of parents and students | 30 | *"Conduct a project to improve habits for students and parents"* |
| | | *"Raise awareness of students and teachers"* |
| | | *"A school for parents, but many parents do not understand"* |
| | | *"That parents apply the school rules at home"* |
| | | *"It is possible to conduct talks but parents usually do not come"* |
| | | *"It is very difficult to raise awareness with the parents"* |
| There is no supervision of compliance with the AGREEMENT in schools | 17 | *"That the authority makes decisions related to the AGREEMENT and disseminates the regulations for teachers, school authorities and parents"* |
| | | *"Applying the AGREEMENT just as it is"* |
| | | *"Making it a mandatory regulation, for example, that there are no school canteens or to remove the food that does not comply"* |
| | | *"A lady sells junk food outside of the school passing it through a hole in the wall. We already reported it to SEP but they do not respond"* |
| | | *"To permit a revision filter for food at the school entrance"* |
| | | *"That there are regular visits from the authorities and training"* |
| | | *"To conduct inspections of what the school canteen sells and apply penalties if they do not comply"* |
| Cultural patterns do not favor healthy eating at home | 10 | *"Change the culture, everything the they are accustomed to eat"* |
| | | *"For the children to pay attention, education starts at home", "It is something cultural"* |
| | | *"To instill the culture of healthy foods"* |
| | | *"The economy is the principal problem for parents in this school"* |
| There is no dissemination of the contents of the AGREEMENT | 8 | *"To disseminate information about the AGREEMENT"* |
| | | *"Use the media to inform"* |
| | | *"To conduct continuous promotion of healthy eating behaviors"* |

*(Continued)*

**Table 3.** (Continued)

| Theme | No. of mentions | Quotes about possible solutions |
|---|---|---|
| Street food vendors and other small businesses sell food prohibited under the AGREEMENT to the children | 8 | *"There are small businesses outside the school where most of the students buy food"* |
| | | *"Outside the school there is a taco restaurant and also a neighbor sells food"* |
| | | *"The policy is not consistent with the permissions that the municipal council provides to street food vendors"* |
| Students do not have enough money to buy healthy food | 6 | *"Healthy foods are more expensive and families do not have enough money"* |
| | | *"The school canteen cannot sell healthy food because it is not affordable for students"* |
| | | *"The school has deficiencies, the students are from a low-income strata"* |
| | | *"The school is in a marginalized zone, there are not enough children to sustain a school canteen, students have to buy from street food vendors"* |

SEP–Ministry of Public Education.

parents could be facilitators rather than barriers to its implementation. Parents were more likely to be reported as facilitators for private schools than for public schools. Of the authorities that selected teachers and school authorities as a facilitator, the most common reasons were "they have/could have an interest in the AGREEMENT" (95.8%, 95%CI 90.4–98.6) and "they are/could be available to disseminate information about the AGREEMENT" (95.8%, 95%CI 90.4–98.6), followed by "they are/could be available to give or receive training related to the AGREEMENT" (83.1%, 95%CI 75.0–89.3).

## Evaluation of the physical environment

On inspection of the school, water fountains were observed in 93.3% of schools (n = 111) (Table 4). However, in only 16.2% of these 111 schools were all the water fountains both clean

**Table 4. Results of the evaluation of the physical environment (n = 119).**

| Indicator | Prevalence % (95% CI) | |
|---|---|---|
| Schools where data collectors observed the existence of water fountains or water dispensers | 93.3 | (87.2–97.1) |
| Schools where all the water fountains were functional* | 20.7 | (13.6–29.5) |
| Schools where all the water fountains were clean* | 18.0 | (11.4–26.4) |
| Schools where all the water fountains were functional and clean* | 16.2 | (9.9–24.4) |
| Schools with advertising of processed foods and/or beverages | 16.0 | (9.9–23.8) |
| Schools with publications referring to the practices promoted by the AGREEMENT, e.g. healthy food habits | 12.6 | (7.2–19.9) |

CI–confidence interval.

*Only schools with water fountains (n = 111).

Note: data was collected by the data collectors through observation. The data collectors did not enter the classrooms.

and functional. An analysis of the actual number of water fountains observed in the schools showed that there was an average of 37 children per water fountain. The mean number of water fountains per school was 10.8 (95%CI 8.7–13.1), of which 43.7% (95%CI 36.8–50.6) of the water fountains observed were functional and 23.4% (95%CI 16.1–30.7) were clean. It is important to mention that some schools have water dispensers inside the classrooms, but it was not possible to verify this–thus the numbers reported in Table 4 may be an underestimation.

In relation to promotional material within the school, more advertisements of SSBs and EDNP foods were observed than publications referring to the practices promoted by the AGREEMENT, e.g. healthy food habits (Table 4). This publicity was located in places such as the school canteen, the walls and the playground.

## Discussion

The 2014 AGREEMENT that establishes the general guidelines for the sale and distribution of prepared and processed foods and beverages in schools of the Mexican National Education System was published over six years ago [13]. However, data collected in a random sample of schools in Hermosillo, Sonora between November 2018 and April 2019 show that its implementation in schools is limited. Only 15.1% (95%CI 9.2–22.8) of the interviewed school authorities considered that their school had fully implemented the AGREEMENT and only 1% (95%CI 0–5.3) of the school canteens were fully compliant with the requirements of the AGREEMENT.

This study of the extent of implementation of a national school food policy is one of very few conducted in low and middle-income countries [32, 33]. Further, apart from New Zealand [21, 22] and Fiji (not yet published), this study from Mexico is the only one completed using the food provision module of INFORMAS in schools. While systematic review evidence suggests that, when implemented, school food policies are generally effective in improving the food environment and dietary intake of school students [10], they are often poorly implemented under real-life conditions [24, 33]. The current study suggests that the current Mexican policy is no exception.

There are several issues that seem to be impacting on the implementation of the AGREEMENT in Mexican schools. The most significant is the fact that the policy is not obligatory and is not being enforced with penalties for breaches, which is a likely contributor to its lax implementation [34]. Interference by the food industry in public nutrition policy is not unusual in Mexico and is likely to have contributed to the policy not being made mandatory [35], as seen with the previous version of the front of pack labelling laws [36]. A concerted effort is needed by academia and civil society, with support from international agencies and foundations, to convert this guideline into a law that would be more likely to have impact and to support its implementation. The Mexican experience with the introduction of the new front of pack warning labels in October 2020 shows that it is possible.

Other issues impacting on the implementation of the AGREEMENT include poor dissemination of, and formal training in, the AGREEMENT. In turn, teachers and parents are receiving only limited (or no) information from the school authorities about what the AGREEMENT establishes and what their responsibilities are under the AGREEMENT. Conveniently, responsibility for its implementation is assigned to the school authorities with help from the Committee of School Food Consumption Establishments (made up of parents and guardians) but neither appear to have been given the necessary tools and support to undertake this responsibility.

While the AGREEMENT requires that the education authorities (Ministry of Education) ensure that the children have access to drinkable water, less than half of the water fountains

observed were actually functional. This proportion is higher than that found by El Poder del Consumidor (11%) in nine schools in the center of Mexico [14] but still well below that required by the AGREEMENT. Further, in some schools that did have water fountains, the school authority interviewed commented that they did not have a filtration system to make the water drinkable, had insufficient water pressure or missing water pipes. A few commented that this was due to robberies and others that some of the water fountains left by previous governments had never been properly installed.

Another serious issue is the fact that the State Ministry of Education seems to be using secondary documents (which are more flexible) rather than the AGREEMENT itself for both canteen inspections and training of the canteen proprietors. Data collectors were able to view some of these secondary documents during the interview with the school authorities when they asked to see a copy of the AGREEMENT. Further, the general opinion of the school authorities is that the verification visits do not result in actions when foods are identified that should not be sold. Clearly, more work is required by the Ministry of Education of Sonora to ensure that the canteen proprietors are properly trained and monitored in accordance with the requirements of the AGREEMENT.

Insufficient funding of schools is another factor that could be affecting the implementation of the AGREEMENT. According to the school authorities, each school (with a school canteen in which the manager was assigned by the Ministry of Education) receives approximately 7 to 10 Mexican pesos per student (approximately 0.50 USD). For public schools, the income provided by school canteens represents their only monthly income for all school necessities. Thus, the education authorities may be reluctant to make changes that affect the profitability or survival of the canteen, e.g. by removing EDNP foods and SSBs from sale, a point also noted by "El Poder del Consumidor" as a finding of their study [14]. Unless schools receive sufficient government funding to cover all essential expenses and that is not linked to canteen sales, it is unlikely that this situation will change.

The finding that there are still mobile food vendors outside of the schools, and that 60.5% of the school authorities report that children regularly buy foods and drinks from them, may make school canteen proprietors reluctant to remove offerings of unhealthy foods. It is also general knowledge that, while the local government may not issue new licenses for mobile food vendors within 100 meters of schools, they do renew existing licenses. Clearly more work is needed in relation to this aspect.

Intervention is needed by the Ministries of Health and Education to improve implementation of the AGREEMENT, including appropriate monitoring and follow-up of compliance. This could be informed by further qualitative research to explore the reasons for more flexible requirements in secondary technical documents as well as to identify any issues in the process and outcomes of the verification visits to schools. Schools also require further training and support to facilitate implementation of the AGREEMENT. They should also be involved in decisions relating to their school canteen as school principals noted that they had no participation in the choice of operator for their school canteen and thus, felt unable to monitor or influence its day-to-day operation.

Among the strengths of this study are that it includes a large, representative sample of all primary schools in Hermosillo; and had an excellent response rate, which minimizes the risk of selection bias. Further, the data collectors used direct observation of the school canteens and school yard to measure compliance rather than relying on self-reported data, which would be more likely to lead to an overestimation of compliance. The tools developed for this study could be used at a larger scale to determine the level of implementation of, and compliance with, the AGREEMENT at a national level. These measures could also serve as a baseline to inform the development and evaluation of the effect of interventions designed to improve

implementation of the AGREEMENT. Finally, the tools can be used for ongoing monitoring to drive increased accountability of government and the private sector for improving the school food environment [37].

A limitation of the study was that only schools in Hermosillo could be included in the sample due to time and resource constraints. However, we expect that similar results would be found in other parts of Sonora and in other states, given the results of the only other evaluation of the 2014 AGREEMENT [14], as well as previous evaluations of the 2010 version [9]. Another limitation of the study was a lower response rate in private schools, despite frequent follow-up from the data collectors. Further, it was not possible to observe mobile food vendors outside of the schools due to limitations in human resources.

## Conclusion

The 2014 AGREEMENT is the principal guideline in Mexico that regulates the school food environment. However, the level of implementation of, and compliance with, the AGREEMENT is very low in elementary schools in Hermosillo, Sonora. This is likely due to the fact that it is an agreement rather than a law and to the lack of penalties for non-compliance. A wide variety of processed foods and beverages are available in school canteens and some type of SSB was found for sale in 100% of them. The availability of drinkable water is not sufficient for all students. In addition, children regularly purchase foods and drinks from mobile food vendors outside of schools that sell foods and beverages that are not allowed in the AGREEMENT. Further, knowledge of the AGREEMENT by the school authorities is limited and in less than a quarter of the schools have they received formal training regarding its contents. Although parents (and students) were identified as the main barriers to the implementation of the AGREEMENT in elementary schools, they are not generally receiving information related to its contents. Thus, further work is required to better support the full implementation of, and compliance with, the AGREEMENT of 2014.

## Supporting information

**S1 Checklist.**
(DOC)

**S1 File. Data collection instruments (Spanish versions).** General data, Interview with school authorities, School canteen instrument, Breakfast menu instrument, Evaluation of the physical environment, Non-participation survey, and Checklist.
(PDF)

**S2 File. Data collection instruments (English versions).** General data, Interview with school authorities, School canteen instrument, Breakfast menu instrument, Evaluation of the physical environment, Non-participation survey, and Checklist.
(PDF)

**S1 Table. Foods and beverages available in school canteens.** List of food and drinks and their prevalence in school canteens.
(PDF)

**S2 Table. Foods and beverages included in school breakfast menus.** List of food and drinks and their prevalence in school breakfast menus.
(PDF)

## Acknowledgments

We thank the Ministry of Education in Sonora for facilitating the entry of the researchers to the selected schools for data collection. We thank the school authorities for their support of, and participation in, this research.

## Author Contributions

**Conceptualization:** Yazmín Hugues, Michelle M. Haby.

**Formal analysis:** Yazmín Hugues.

**Investigation:** Yazmín Hugues.

**Methodology:** Yazmín Hugues, Rolando G. Díaz-Zavala, Trinidad Quizán-Plata, Camila Corvalán, Michelle M. Haby.

**Supervision:** Michelle M. Haby.

**Visualization:** Rolando G. Díaz-Zavala, Trinidad Quizán-Plata, Camila Corvalán.

**Writing – original draft:** Yazmín Hugues.

**Writing – review & editing:** Yazmín Hugues, Rolando G. Díaz-Zavala, Trinidad Quizán-Plata, Camila Corvalán, Michelle M. Haby.

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
