## [Decision Letter · Decision Letter 0]

29 Mar 2021

PONE-D-20-39852

Poor compliance with school food environment guidelines in elementary schools in Northwest Mexico: a cross-sectional study

PLOS ONE

Dear Dr. Haby,

Thank you for submitting your manuscript to PLOS ONE. After careful consideration, we feel that it has merit but does not fully meet PLOS ONE’s publication criteria as it currently stands. Therefore, we invite you to submit a revised version of the manuscript that addresses the points raised during the review process.

Both the qualified reviewers provided comments but their recommendations are opposite. Reviewer 1 recommended rejection because your paper is a policy report not a research article, while Reviewer 2 recommended acceptance. I have read your paper carefully and I agree with Reviewer 1 that your method is very simple and descriptive. However, your research design, method, and findings did answer the question you aim to study: what % of schools that implemented the policy or complied with the policy. In this sense, I think your research has met the requirement of PLOS ONE publication criteria. Therefore, I invite you to revise your paper based on Reviewer 1’s comments. You don’t have to follow all the suggestions as some of them require data or work beyond your current survey design; try to address those concerns as much as you can.

We look forward to receiving your revised manuscript.

Kind regards,

Shihe Fu, Ph.D.

Academic Editor

PLOS ONE

Journal Requirements:

4. Please include copies of the questionnaires and data collection tools, in English, as Supporting Information, or include a citation if they have been published previously.

5. Thank you for stating the following in the Funding Section of your manuscript:

[The Division of Biological and Health Sciences of the University of Sonora funded the 574printing of the instruments for data collection as well as fuel costs. YHA received a Master’s 575degree scholarship from CONACYT (National Council of Science and Technology.]

 [The funders had no role in study design, data collection and analysis, decision to publish, or preparation of the manuscript.]

7. Please include captions for all your Supporting Information files at the end of your manuscript, and update any in-text citations to match accordingly. Please see our Supporting Information guidelines for more information: http://journals.plos.org/plosone/s/supporting-information.

Reviewers' comments:

Reviewer's Responses to Questions

**Comments to the Author**

1. Is the manuscript technically sound, and do the data support the conclusions?

Reviewer #1: No

Reviewer #2: Yes

2. Has the statistical analysis been performed appropriately and rigorously? 

Reviewer #1: No

Reviewer #2: Yes

3. Have the authors made all data underlying the findings in their manuscript fully available?

Reviewer #1: Yes

Reviewer #2: Yes

4. Is the manuscript presented in an intelligible fashion and written in standard English?

Reviewer #1: Yes

Reviewer #2: Yes

5. Review Comments to the Author

Reviewer #1: Referee report for Manuscript “PONE-D-20-39852”

This paper, entitled “Poor compliance with school food environment guidelines in elementary schools in Northwest Mexico: a cross-sectional study”, reports a survey result. The survey is conducted to collect data on the implementation of Federal Government guideline for the sale and distribution of food and beverage (AGREEMENT) in the context of high overweight and obese prevalence in Mexican. While the paper raises important questions, it has the following flaws:

1. it is a report, not a research article. It only reports the very low rate of compliance rates without analyzing the behind reasons. It could be extended, for example, by analyzing the spillover effect of mobile food vendors on the street on the provision of unhealthy (sugar-sweetened beverages) foods of public schools, or by examining the difference between public and private schools. Another way to extend it is to examine why government seems use secondary document, rather than the AGREEMENT, to inspect schools. Are there any corruptions?

2. The contribution of this paper to literature is quite limited. This paper brings us updated survey results, with findings similar to the previous literature.

Reviewer #2: Using a random sample of schools in Hermosillo, Sonora, this study shows the poor compliance of 2014 AGREEMENT, which establishes the general guidelines for the sale and distribution of foods and beverages in schools in Mexico. Rather than relying on self-reported data, this study uses direct observation of the school canteens and school yard to measure compliance. These measures are performed to a high technical standard. The study further discusses the barriers and facilitators to implementation of the AGREEMENT in detail.

6. PLOS authors have the option to publish the peer review history of their article (what does this mean?). If published, this will include your full peer review and any attached files.

Reviewer #1: No

Reviewer #2: No

---

## [Author Response · Author response to Decision Letter 0]

13 May 2021

Response to reviewer comments:

Our responses follow each comment.

Reviewer #1: Referee report for Manuscript “PONE-D-20-39852”

This paper, entitled “Poor compliance with school food environment guidelines in elementary schools in Northwest Mexico: a cross-sectional study”, reports a survey result. The survey is conducted to collect data on the implementation of Federal Government guideline for the sale and distribution of food and beverage (AGREEMENT) in the context of high overweight and obese prevalence in Mexican. While the paper raises important questions, it has the following flaws:

1. it is a report, not a research article. It only reports the very low rate of compliance rates without analyzing the behind reasons. It could be extended, for example, by analyzing the spillover effect of mobile food vendors on the street on the provision of unhealthy (sugar-sweetened beverages) foods of public schools, or by examining the difference between public and private schools. Another way to extend it is to examine why government seems use secondary document, rather than the AGREEMENT, to inspect schools. Are there any corruptions?

Response:

We disagree with the reviewer on this aspect. While the study is a descriptive cross-sectional study this is a valid epidemiological research design, and the study offers important and high-quality results (as noted by reviewer #2). This is the first study of a large, random sample of schools in Mexico, that has used direct observation of the school canteens and school yard to measure compliance rather than relying on self-reported data (which is more likely to lead to an overestimation of compliance). The observation method used followed a standardized protocol, is replicable and valid and therefore provides the opportunity to monitor progress. It was also a necessary first step to show that compliance was low and to highlight possible reasons for this, such as lack of training of the school authorities. 

Further, while the objective of the study was not to analyze the reasons for low compliance, the discussion raises several potential explanations that may serve as hypotheses for future studies. 

2. The contribution of this paper to literature is quite limited. This paper brings us updated survey results, with findings similar to the previous literature.

Response:

We disagree that the findings are similar to the previous literature. The findings from this study, using direct observation, are much lower than previous studies in Mexico that used self-report. And, as mentioned in point 2, it will serve to show that there is indeed a problem with the implementation of the guidelines so that steps can be taken to improve both the guidelines themselves and their implementation. An additional strength of this type of study is that, the monitoring of food environments can help drive increased accountability of governments and the private sector and stimulate policy changes to improve them (Sacks et al. 2021). A comment has been added to the discussion (page 26) to this effect.

Sacks G, Kwon J, Vandevijvere S, Swinburn B. Benchmarking as a public health strategy for creating healthy food environments: an evaluation of the INFORMAS initiative (2012–2020). Annu Rev Public Health. 2021;42(1):345-62.

Reviewer #2: Using a random sample of schools in Hermosillo, Sonora, this study shows the poor compliance of 2014 AGREEMENT, which establishes the general guidelines for the sale and distribution of foods and beverages in schools in Mexico. Rather than relying on self-reported data, this study uses direct observation of the school canteens and school yard to measure compliance. These measures are performed to a high technical standard. The study further discusses the barriers and facilitators to implementation of the AGREEMENT in detail.

Response:

We thank you for your comments and agree that the use of direct observation was a strength of the study, as well as the high technical standards that we employed to ensure valid results.

---

## [Decision Letter · Decision Letter 1]

15 Jul 2021

PONE-D-20-39852R1

Poor compliance with school food environment guidelines in elementary schools in Northwest Mexico: a cross-sectional study

PLOS ONE

Dear Dr. Haby,

Thank you for submitting your manuscript to PLOS ONE. After careful consideration, we feel that it has merit but does not fully meet PLOS ONE’s publication criteria as it currently stands. Therefore, we invite you to submit a revised version of the manuscript that addresses the points raised during the review process.

Your paper is close to be publishable. The second reviewer recommended acceptance and the first reviewer provided some comments for a minor revision. Please try to address those comments as much as you can. I will not send out the revised version for another round review; instead, I will make the final decision based on my careful reading of your new version.

We look forward to receiving your revised manuscript.

Kind regards,

Shihe Fu, Ph.D.

Academic Editor

PLOS ONE

Journal Requirements:

Reviewers' comments:

Reviewer's Responses to Questions

**Comments to the Author**

1. If the authors have adequately addressed your comments raised in a previous round of review and you feel that this manuscript is now acceptable for publication, you may indicate that here to bypass the “Comments to the Author” section, enter your conflict of interest statement in the “Confidential to Editor” section, and submit your "Accept" recommendation.

Reviewer #1: (No Response)

Reviewer #2: All comments have been addressed

2. Is the manuscript technically sound, and do the data support the conclusions?

Reviewer #1: (No Response)

Reviewer #2: Yes

3. Has the statistical analysis been performed appropriately and rigorously? 

Reviewer #1: (No Response)

Reviewer #2: Yes

4. Have the authors made all data underlying the findings in their manuscript fully available?

Reviewer #1: (No Response)

Reviewer #2: Yes

5. Is the manuscript presented in an intelligible fashion and written in standard English?

Reviewer #1: (No Response)

Reviewer #2: Yes

6. Review Comments to the Author

Reviewer #1: (No Response)

Reviewer #2: Comparing to the previous literature which mainly rely on self-reported data, this study uses direct observation of the school canteens and school yard to measure compliance. It is important to report the basic facts of compliance in this rigorous way.

7. PLOS authors have the option to publish the peer review history of their article (what does this mean?). If published, this will include your full peer review and any attached files.

Reviewer #1: No

Reviewer #2: No

---

## [Author Response · Author response to Decision Letter 1]

20 Sep 2021

Response to the reviewer

1. The authors could report implementation and compliance rate by public schools and private schools, and by principal/teachers (administrative staff), to enrich their results. In doing so, the authors can help readers to further understand whether the implementation and compliance rates are different in public/private schools and etc.

Thank you for the suggestion. We investigated whether the implementation rate was higher in private schools than public schools and found that it was. Full implementation of the agreement was more likely to be reported by private schools than public schools (33.3 vs 9.0%, p=0.009, Fisher’s exact test). We have added this finding to the manuscript (lines 338-340). In relation to compliance, the total number of schools that are partially or fully complying with the agreement is too small (5/103) to enable a meaningful comparison between private and public schools. We are unsure what comparison the reviewer is alluding to in relation to principal/teachers (administrative staff). 

2. The school authorities will always think that parents and students are barriers, while they are the facilitators to implantation of the AGGEEMENT. Therefore, figure 2 does not tell us new information. The authors should consider report the difference of reasons (such as lack of knowledge) between public and private schools. In doing so, the authors can help readers get new information about whether there are different reasons of poor compliance for different types of school.

Analysis of the reasons why the different groups/persons are barriers is complicated by the fact that the question reporting the reasons is not specific to each of the groups – thus the interpretation of any differences in the reasons is difficult. When we analyzed the differences between public and private schools regarding the reasons (without matching to the group/person), we only found a difference between public and private schools for ‘Lack of interest’. No differences were found for the other reasons. Thus, we have not added this information to the paper. Further, we think that the analysis/results shown in Table 3 is a more useful way to present this information. However, in response to the reviewer’s comment we have replaced figure 2 with a figure that compares the barriers and facilitators to implementation for public and private schools. The graph of barriers shows that parents, students and vendors were more likely to be reported as barriers for public schools than for private schools, which we have now noted in lines 380-382). And the graph of facilitators shows that parents were more likely to be reported as facilitators for private schools than for public schools (see lines 403-404).

---

## [Editor Report · Decision Letter 2]

26 Oct 2021

Poor compliance with school food environment guidelines in elementary schools in Northwest Mexico: a cross-sectional study

PONE-D-20-39852R2

Dear Dr. Haby,

We’re pleased to inform you that your manuscript has been judged scientifically suitable for publication and will be formally accepted for publication once it meets all outstanding technical requirements.

Kind regards,

Shihe Fu, Ph.D.

Academic Editor

PLOS ONE
---

## [Editor Report · Acceptance letter]

2 Nov 2021

PONE-D-20-39852R2 

Poor compliance with school food environment guidelines in elementary schools in Northwest Mexico: a cross-sectional study 

Dear Dr. Haby:

I'm pleased to inform you that your manuscript has been deemed suitable for publication in PLOS ONE. Congratulations! Your manuscript is now with our production department. 

Kind regards, 

on behalf of

Dr. Shihe Fu 

Academic Editor

PLOS ONE